# Contact Dermatitis to Diabetes Medical Devices

**DOI:** 10.3390/ijms241310697

**Published:** 2023-06-27

**Authors:** Mikołaj Cichoń, Magdalena Trzeciak, Małgorzata Sokołowska-Wojdyło, Roman J. Nowicki

**Affiliations:** Department of Dermatology, Venereology and Allergology, Medical University of Gdansk, 80-214 Gdansk, Poland; mikolaj.cichon@gumed.edu.pl (M.C.); malgorzata.sokolowska-wojdylo@gumed.edu.pl (M.S.-W.); rnowicki@gumed.edu.pl (R.J.N.)

**Keywords:** allergic contact dermatitis, irritant contact dermatitis, diabetes medical devices, glucose sensors, insulin pumps, isobornyl acrylate, IBOA, 2,2′-methylenebis(6-tert-butyl-4-methylphenol) monoacrylate, MBPA

## Abstract

Skin adverse reactions to diabetes medical devices have been reported frequently over recent years. Adhesives attaching glucose sensors and continuous insulin infusion sets to the skin are proven to cause both allergic contact dermatitis and irritant contact dermatitis in patients with diabetes mellitus. Several allergens contained in adhesives and/or parts of medical devices are documented to cause allergic contact dermatitis, with acrylate chemicals being the most common culprit-especially isobornyl acrylate (IBOA), but also 2,2′-methylenebis(6-tert-butyl-4-methylphenol) monoacrylate or cyanoacrylates. Epoxy resin, colophonium and nickel were also identified as causative allergens. However, repetitive occlusion, maceration of the skin and resulting disruption of the skin barrier seem to have an impact on the development of skin lesions as well. The purpose of this study is to highlight the burden of contact dermatitis triggered by diabetes medical devices and to show possible mechanisms responsible for the development of contact dermatitis in a group of diabetic patients.

## 1. Introduction

The management of diabetes mellitus (DM) has been vastly improved due to the broader access to technological devices such as glucose sensors (GS) and continuous subcutaneous insulin infusion (CSII) sets. The two main categories of GS are flash glucose monitoring (FGM) and continuous glucose monitoring (CGM). FGMs do not require calibration with self-measurements of glucose levels from finger prick tests, but users need to scan the sensor manually. On the other hand, CGMs provide real-time tracking of interstitial glucose levels, though some of them still require calibration [1]. Avoiding short-term complications of DM (such as diabetic ketoacidosis or frequent hypoglycemia episodes) and long-term complications (such as retinopathy, nephropathy or neuropathy) is essential and more accurate glycemic control facilitated by early use of GS and CSII by minimizes these risks. Use of medical devices also increases patients’ compliance with medication and enhances their quality of life [2]. Although the management of DM1 has been revolutionized by technological achievements, the incidence rate of dermatological complications resulting from using diabetes medical devices is increasing. The incidence of DM type 1 and DM type 2 continues to rise in Europe [3]. Over recent years, cases of irritant contact dermatitis (ICD) and allergic contact dermatitis (ACD) caused by diabetes medical devices have been reported, with acrylates being the most common culprit in the latter [4]. Apart from contact dermatitis, other cutaneous adverse effects frequently seen include skin infections, unspecified skin eruptions, urticaria or oedema [5]. Berg et al. report that almost 90% of patients using CSII experienced adverse skin effects with non-specific eczema being present most frequently in 25.7% of patients [6]. Since skin adverse effects remain the most common reason for discontinuation of CGM, it is crucial to determine the role of allergens in triggering contact dermatitis and the mechanisms beyond it [7]. This article focuses on allergens eliciting ACD as well as on the factors leading to ICD in diabetic patients.

### Materials and Methods

A review of the literature concerning the problem of contact dermatitis in patients with diabetes who use medical devices was conducted using PubMed and Web of Science between March 2023 and June 2023. No restrictions were placed on article types, publication date, country, journal or publisher. The following terms were searched for: diabetes medical devices, diabetes devices, insulin infusion sets, insulin pumps, continuous glucose monitoring system, flash glucose monitoring system, glucose monitoring system, glucose sensors, acrylate, acrylates, isobornyl acrylate, IBOA and contact dermatitis, allergic contact dermatitis and irritant contact dermatitis. A backward search (scanning the references included in relevant articles), as well as forward search (search for relevant articles in which the original article was cited after being published), was also conducted. Since the clinical problem of contact dermatitis elicited by diabetes medical devices is not that widely explored and covered in the literature yet and many studies are currently being performed, personal observations were also included (these are appropriately indicated in the text).

## 2. Contact Dermatitis Triggered by Diabetes Medical Devices

Contact dermatitis (CD) is a very common inflammatory skin disease characterized by pruritic, eczematous-scaling lesions sometimes accompanied by vesicles or impetiginization. The two main types of contact dermatitis are ACD and ICD, whose clinical pictures in most cases are indistinguishable. ICD is more frequent and stands for approximately 80% of all contact dermatitis cases [8], whereas the prevalence of ACD is as much as 20% [9]. Statistically, contact dermatitis affects women twice as often as men with the onset of symptoms between 12 and 16 years of age in 15% [10,11]. Notably, the coexistence of ACD and ICD at the same time is possible. Since DM1 accounts for about 85% of all diabetes cases in patients under the age of 20, with a peak between 10 and 14 years old, most device users are children and adolescents [12]. Therefore, dermatological problems that stem from the use of either GS or CSII sets are most common in the younger group of patients.

Whilst several report studies and case series have been published across the past few years, there is a paucity of reliable systematic reviews or meta-analysis presenting the actual incidence rate of ACD and ICD triggered by diabetes medical devices. Nonetheless, contact dermatitis elicited by the use of insulin pumps and/or glucose sensors can be either allergic or irritant [13]. 

### 2.1. Risk Factors

There are risk factors for the development of local skin reactions to diabetes medical devices. One of them is the use of diabetes devices in the past, which could contain allergens (such as IBOA), regardless of the presence of skin lesions. Svedman et al. pinpointed that patients sensitized through the use of one medical device are not free from future episodes of ACD when using another product [14]. Another risk factor is atopic dermatitis history and consequent epidermal barrier disruption or other epidermal barrier disorders (e.g., fillagrin deficiency). Such conditions facilitate the penetration of allergens through the skin barrier leading to more prompt sensitization. Furthermore, patients with skin barrier abnormalities have lower inflammatory thresholds for external irritant factors and are more likely to develop ICD [15].

### 2.2. Allergic Contact Dermatitis

ACD is an example of a type IV hypersensitivity reaction according to Coombs and Gell classification, in which dermal dendritic cells (DCs) and epidermal DCs (Langerhans cells) play a key role in sensitization and elicitation phases. During the sensitization process, an individual is first exposed to an allergen (hapten), which reacts with DCs and, as a hapten-peptide complex, migrates to regional lymph nodes of the skin and prime naïve Th-cells. DCs present the haptens on their major histocompatibility complex molecules (MHC) to antigen-specific T-cell receptors (TCRs) leading to the formation of hapten-specific memory and effector T-cells. Upon re-exposure, the hapten is recognized by already sensitized hapten-specific T-cells, which migrate to the skin. Following this, hapten-specific cytotoxic CD8+ T lymphocytes, via proinflammatory cytokine cascade, elicit skin lesions that are clinically seen as ACD [16,17]. DCs, by priming the naïve T-cells, act as a link between the innate and adaptive immune system. (Meth)acrylates are suspected to be the major contact allergens found in medical adhesives.

What is currently being investigated is why not all individuals exposed to an allergen become sensitized towards it and, consequently, might develop ACD. Genetic tendencies and environmental factors seem to predispose certain groups and put them at higher risk of developing ACD [18]. Family members tend to develop ACD more frequently, which suggests the role of genetics but also pinpoints environment and ethnicity as risk factors [18]. Mutations in the genes encoding proteins of the epidermal skin barrier (e.g., filaggrin) [19] or genetically influenced polymorphism for enzymatic activities (e.g., higher N-acetyltransferase activity is linked with contact dermatitis) play a role as well [20,21]. Gene polymorphisms in coding regions of enzymes such as glutathione S-transferases M1 and T1 [22] or angiotensin-converting inhibitors [23] are associated with an increased risk for ACD. Cytokine gene polymorphisms for promoters for tumor necrosis factor alpha [24] or interleukin 16 [25] are genetic risk factors directly connected to the immunological response.

#### 2.2.1. Patch Testing

The gold standard in the diagnosis of ACD is patch testing. When suspecting ACD connected to diabetes medical devices, acrylate-series patch tests including IBOA should be performed. From our observations, acrylate allergens very often do not give positive results until the last reading taken after 7 days, so it is crucial to perform all three readings after 48 h, 72 h and 7 days. The difficulties of patch testing and diagnosis of cases with contact dermatitis from medical devices have been discussed by Ulriksdotter et al. as the role of an untested allergen in the development of dermatitis must be kept in mind. Frequently, the causative allergen is not identified [26]. Cases with negative patch tests could be described as possible ACD or unspecified contact dermatitis.

The question arises whether diabetic patients should undergo patch testing prior to using the device. There is, however, no unequivocal answer. This difficult topic is not covered in any guidelines and should be treated in a patient-oriented way. The decision on performing patch tests (or not) is strongly dependent on the clinical picture and symptoms patients present. If a patient has a history of contact allergy to glues, sealants, adhesives, etc., it seems reasonable to extend diagnostics with patch tests as possible results towards epoxy resin, colophony or isobornyl acrylate can help with choosing the appropriate device free from these allergens, thus preventing possible elicitation of ACD. On the other hand, exposing patients to new allergens during patch tests and possible sensitization, even if they do not present clinical symptoms of contact allergy towards any allergens, remains highly questionable. The identification of the causative allergen(s) in patients with contact allergy towards medical devices is immensely challenging, sometimes resembling the ‘trial and error’ method. Finally, not every patient using insulin pumps and/or glucose sensors eventually elicits contact dermatitis. Unfortunately, we have not found any official data describing the percentage of patients with DM experiencing contact dermatitis from the medical devices they use. From our observations, this number fluctuates by a few percent, though the trend is upward in recent years (personal observations M.C. and M.T.).

In the following paragraphs, we outline the contact allergens eliciting ACD and evaluate the factors contributing to ICD in the users of diabetic medical devices.

#### 2.2.2. Isobornyl Acrylate

Acrylates are created by polymerization of monomers derived from (meth)acrylic acid. Acrylic monomers are proven to cause the most documented cutaneous reactions. In contrast, acrylic polymers are relatively inert, though every polymerized acrylate very often contains trace amounts of residual monomers [27]. It has been shown that all types of acrylates have the potential to sensitize, with monoacrylates being considered weaker sensitizers and multifunctional acrylates as stronger allergens [27].

Isobornyl acrylate (IBOA; CAS 5888-33-5), the 2020 American Contact Dermatitis Society Allergen of the Year, is a liquid and reactive acrylate monomer widely used in plastic material and ink manufacture. In everyday life, IBOA can be found in glues, adhesives, resins, inks and solvents, in which it offers good resilience, flexibility and hardness [28]. Therefore, it is also a perfect compound for the manufacture of adhesives used to attach GS and CSII sets to the skin. On safety sheets, it is classified as an irritant substance. As Foti et al. report, IBOA may sometimes play the role of a hidden allergen collected as an impurity during the industrial process [29]. In the past, IBOA, though repeatedly identified in occupational components, was rarely the cause of ACD and remained in the shadow of other acrylates responsible for ACD such as 2-hydroxyethyl methacrylate or ethylene glycol dimethacrylate [30,31]. The first reports of IBOA-induced ACD in diabetic patients are from 1995 when two young women experienced eczema at the sites of insulin pump attachment [32]. IBOA was one of the allergens detected in the UV-cured (ultraviolet-cured) glue used to fix the needle into the plastic set. Both patients had positive patch test results for this acrylate. In 2016, the accidental presence of IBOA in the FreeStyle Libre sensor (FreeStyle Libre; Abbott Diabetes Care, Alameda, CA, USA) was established by a group of Belgian dermatologists [28]. From this moment onward, more and more diabetic patients with similar skin lesions were patch tested towards IBOA, proving this acrylate to be the culprit in many cases. In Finland, 81% of the patients experiencing adverse skin reactions to FreeStyle Libre were sensitized towards IBOA [33,34]. In the years since, it has been confirmed that, apart from FreeStyle Libre, IBOA is a contact allergen detectable in: (i) the housing [35] (1.11 ± 0.12 μg/mL), adhesive [35] (0.26 μg/mL) and in the Enlite sensor itself [36] (10 μg/sensor) (Medtronic, Fridley, MN, USA); (ii) a tubeless insulin pump Omnipod [37] (5 μg/patch and 190 μg/sensor) (Insulet Corporation, Billerica, MA, USA); (iii) insulin infusions sets Paradigm MiniMed Quick-Set and Paradigm MiniMed Sure-T [38] (Medtronic, Fridley, MN, USA); (iv) insulin infusion set Accu-Chek Insight Flex (Roche Diabetes Care, Indianapolis, IN, USA) [38]; (v) in all following parts of the Medtrum A6 TouchCare (Medtrum Technologies, Shanghai, China): 1 μg in the sensor, 3 μg in the sensor adhesive patch, 30 μg in the insulin pump reservoir, 6 μg in the reservoir patch adhesive [39]. Initially, gas-chromatography-mass-spectrometry (GC-MS) analysis showed that the related Paradigm Minimed Silhouette infusion does not contain IBOA within detection limits [38], but a recent report of a 15-year-old boy from Poland suggests that the Silhouette set might still contain IBOA in untraceable amounts, but enough to elicit contact dermatitis [40]. Unfortunately, despite the ongoing saga of skin reactions towards diabetes medical devices, it has already been reported that the relatively new insulin pump system YpsoPump (Ypsomed, Burgdorf, Switzerland) also contains IBOA, and the first cases of ACD elicited by this device are known [41]. An alternative for IBOA-sensitized patients was supposed to be the monitoring system Dexcom G6 (Dexcom Inc., San Diego, CA, USA), which was recommended as an IBOA-free device. According to studies performed between 2018 and 2019, IBOA concentrations in the adhesives of Dexcom G5 and Dexcom G6, measured with GC-MS, were below the limit of quantification, which was 0.10 μg/mL for IBOA diluted in methanol [35,42]. In 2020, the modification of the adhesive in Dexcom G6 appeared, and an increasing number of patients started to experience skin problems towards this sensor. The company confirmed that, in order to obtain better skin fixation, an acrylate derivate was exchanged (no precise name of the substance was given), but at the same time it was maintained that the Dexcom G6 system is IBOA-free [43]. Svedman et al. investigated the culprit of ACD in 11 patients using Dexcom G6 and, contrary to the previous findings, identified IBOA in the new ‘IBOA-free’ adhesive patches (0.1–0.7 μg/patch) and in extracts of the sensors (0.8–1.3 μg), whereas extracts from the plastic parts were free from IBOA. The detection limit for IBOA diluted in acetone in this study was 0.01μg/mL. These outcomes, together with positive patch test results, proved IBOA as the contact allergen responsible for the majority of contact allergies in the group of patients using Dexcom G6 [14]. However, most of the patients used IBOA-containing devices (Omnipod insulin pump or Freestyle Libre sensor) prior to switching to Dexcom G6. These case reports pointed out that the issue of switching to ‘allergen-free’ and more useful devices might not always free the patients from future contact reactions. However, the aforementioned studies aiming to detect IBOA in the Dexcom G6 sensor set different analytical limits. In one of them, the limit of quantification (LOQ) for IBOA in methanol was 0.10 μg/mL (no IBOA was detected), whereas the limit of detection (LOD) for IBOA in acetone in another study was 0.01 μg/mL (IBOA detected). The LOQ is the lowest analytical concentration of a substance that can be precisely and accurately measured by an analytical procedure, meeting the usually international acceptance criteria for bias or imprecision [44,45]. However, LOQ and LOD are not equivalent, and cannot be used interchangeably. The major differentiating factor between them is the underlying accuracy and precision. LOQ must always be reported with suitable trueness, reliability and quality criteria, whereas for LOD, no quantification is required. It is designed to show the lowest concentration of a substance in a sample that is greater than zero (the absence of the substance) [46]. If the same limits were set in the studies identifying IBOA in Dexcom G6, the outcomes would be more reliable and, maybe, more cohesive. On the other hand, it is generally acknowledged that even trace amounts of a contact allergen can elicit ACD, so the clinical value of either LOD or LOQ might be disputed. Furthermore, different solvents to dilute IBOA were used (methanol vs. acetone), which might have influenced the final results as well. An actual alternative for IBOA-sensitized patients can be the Eversense XL continuous monitoring system (Roche, Basel, Switzerland), whose sensor is placed underneath the skin in the upper arm, allowing for continuous measures of glucose levels for up to 6 months. In none of the components of the Eversense (implanted sensor, transmitter, two types of adhesive patches) was IBOA found (LOQ < 0.10 μg/mL), making it a viable option for patients with IBOA allergy [47]. Additionally, GC-MS analysis of the new generation FreeStyle Libre 2 sensor did not detect any IBOA residue [48].

#### 2.2.3. Other Acrylate Chemicals

##### 2,2′-Methylenebis(6-*tert*-butyl-4-methylphenol) Monoacrylate (MBPA)

Apart from IBOA, Svedman et al. identified a new allergen in the adhesive of the newer DexcomG6-2.2′-methylenebis(6-*tert*-butyl-4-methylphenol) monoacrylate (MBPA; CAS 61167-58-6), which so far had not been linked with the problem of skin reactions to diabetes medical devices [49]. Therefore, the authors suspect that both IBOA and MBPA could be contact allergens present in Dexcom G6. Further investigation was performed by Oppel et al., who in patients using Dexcom G6 with no previous history of IBOA-sensitization performed patch tests for MBPA in three concentrations (0.1%, 0.3% and 0.5%) and to 2.2′-methylenebis(6-*tert*-butyl-4-methylphenol) (MBP; CAS 119-47-1) also in three concentrations (0.1%, 0.5% and 1.0%) [43]. MBP is a substance related to MBPA but without the acrylate group. In line with previous studies, 0.1% IBOA was patch tested as well. For IBOA and MBP, there were no positive patch tests. Patients reacted to MBPA, with the strongest reaction to 0.5% concentration (no reaction towards MBPA 0.1% was observed). Furthermore, in the same study, MBPA and MBP were detected in the Dexcom G6 new series from 2020 (LOQ MBPA 0.40 μg/mL and LOQ MBP 0.46 μg/mL), while in the series 2018/2019 their presence was not shown. This study showed that MBPA is an actual contact allergen in the new Dexcom G6 series responsible for ACD in patients not sensitized to IBOA [43]. Presumably, MBPA was the acrylate added to support the fixation of Dexcom G6, though the presence of IBOA must be still borne in mind.

##### Dipropylene Glycol Diacrylate (DPGDA)

DPGDA (CAS 57472-68-1) has been previously linked with occupational dermatitis in the painting industry, accounting for 18% of positive patch test results in a group of patients allergic to acrylic monomers [50]. In 2022, three cases of ACD caused by DPGDA in the Omnipod were reported. All patients reacted to 0.1% DPGDA and two of them additionally towards 0.01% concentrations [51]. Though the same authors, detected the presence of IBOA in the Omnipod before, other acrylates, such as DPGDA could not be identified, probably due to less sensitive GC-MS used in the past [37].

##### Cyanoacrylates

Cyanoacrylates (e.g., methyl-2-cyanoacrylate [CAS: 137-05-3] or ethyl-2-cyanoacrylate [CAS: 7085-85-0]), thanks to their ability to polymerize rapidly, can form very strong bonds with metals, plastics, rubbers and biological tissues, and are used mainly in the production of fast-acting glues [27]. In 2016, fabric parts of the Dexcom G4 Platinum containing cyanoacrylates (ethyl-2 cyanoacrylates) were responsible for ACD in diabetic patients [52,53]. Subsequent studies in 2017 confirmed the presence of ethyl cyanoacrylate in this device, a fact that has also been confirmed by the manufacturer [4,54]. In 2020, in the extract of the sensor and the insulin reservoir of Medtrum A6 TouchCare (Medtrum Technologies, China, Shanghai), GC-MS analyses indicated ethyl-2-cyanoacrylate [39].

##### Phenoxypoly(ethylenoxy) Ethylacrylate (PEEA) and β-Carboxyethyl Acrylate (BCA; CAS 24615-84-7)

Along with the first detection of IBOA in a medical device in 1995, other acrylate chemicals proved to be culprits in contact allergy cases, namely PEEA and BCA [32]. BCA was later reported to be one of allergens responsible for an epidemic of occupational dermatitis from acrylic glue amongst Polish workers involved in the production of electric coils for television displays [31]. In 2001, PEEA was also the culprit allergen in ACD in a 38-year-old woman with diabetes treated with an insulin pump [55].

##### N,N-Dimethylacrylamide (DMAA; CAS 2680-03-7)

DMAA is an easily polymerized monomer used as a precursor in the synthesis of hydrogels and polymer coatings and was highlighted to sensitize (0.1% DMAA in PET) and elicit ACD in seven patients using FreeStyle Libre. When analyzed with GC-MS, DMAA was found in the extract of the sensor (≈2 μg/cm^2^) but was not detected in the adhesive sensor patch (<0.5 μg/cm^2^). Six out of seven patients were concomitantly sensitized towards IBOA, which is also present in FreeStyle Libre [56]. DMAA was also shown to be contained in the extracts from the Enlite sensor [36].

#### 2.2.4. Non-Acrylic, Clinically Important Allergens Found in Diabetes Medical Devices

##### Epoxy Resin

Epoxy resin, a well-known cause of ACD, is believed to be the first discovered contact allergen to trigger ACD in two users of an insulin pump (Actrapid autosyringe infusion set). Both patients were positive for epoxy resin (one patient additionally to p-tert-butylphenol-formaldehyde), which the manufacturer was using to bind tubes and needles [57].

##### Colophonium

Colophonium (known as colophony or rosin) is a mixture of >100 compounds derived from pine trees [58]. The exact composition of colophonium varies as it is dependent on the type of pine trees it is derived from, as well as on the extraction and storage techniques. The exact list of allergenic compounds in colophonium is yet to be characterized [59]. Colophony has many uses in industry, but one of them is as a fast-acting adhesive material. Passanisi et al. described two patients, an 8-year-old girl using an Enlite sensor and a 10-year-old boy treated with an Omnipod insulin pump, who experienced eczematous pruritic lesions at application sites. Patch testing revealed that both patients were sensitized towards colophonium 20% in PET. The presence of colophonium in the adhesive on the glucose sensor and in the adhesive on the insulin pump was confirmed by the manufacturers [60]. Colophonium-related substances, such as methyldehydroabietate, were also detected in all extracts acquired from the glucose sensor and the insulin reservoir of Medtrum A6 Touchcare System (Medtrum Technologies, Shanghai, China) [39]. The manufacturer confirmed the device contains up to 15% of modified colophonium. Svedman et al., by observing the reactivity pattern of patch tests performed in a group of diabetic patients, postulated the potential presence of colophonium derivates in Dexcom G6. However, this has not been confirmed and requires further investigation [14].

##### Nickel

Nickel remains the most common allergen that gives positive results in patch tests and can be found in many items and workplaces [17]. There are two reports describing ACD caused by nickel-containing needles in infusion sets [61,62]. However, these cases were reported a long time ago (in 1985 and 1998) and to date, we have not come across other similar reports. Possibly due to the high awareness of its sensitization properties, nickel has been removed from the composition of diabetes medical devices used nowadays. From our experience, the information we received from the manufacturers does confirm the absence of nickel in the devices. Nonetheless, nickel should be considered as a potential culprit and suspected allergen in the group of diabetic patients.

##### 1-Benzoylcyclohexanol

1-Benzoylcyclohexanol is a UV photoinitiator compound contained in UV-cured glue. There is one report of 1-Benzoylcyclohexanol causing ACD in a user of an insulin pump [32].

#### 2.2.5. Cross-Interactions or Co-Reactions?

Sensizitation to methacrylates may induce cross-reactivity to acrylates, but not vice versa [50]. However, studies show that compounds that are not listed on safety data sheets are sometimes still present in commercial products abundant in acrylate chemicals, and this supports the suggestion of possible concomitant positive patch test reactions, rather than cross-reactions [63]. Taking into account insufficient cooperation with the medical devices’ manufacturers, who are hesitant to share the exact composition of all parts of their diabetes devices, we cannot rule out co-reactions in patients patch tested for several acrylates. Though not clearly stated, it is generally believed that IBOA does not cross-interact with other acrylate derivates, a belief that is supported by the statement of occupational dermatologists performing extended series with acrylate patch tests [28]. Nonetheless, none of the available (meth)acrylates in patch test services can be a marker for contact allergy towards IBOA, and every patient with contact dermatitis elicited by diabetes devices should be patch tested for IBOA. There are documented cross-reactions between acrylates (methyl-acrylate and ethyl-acrylate) and dimethyl fumarate (or its isomer dimethyl maleate), though the latter has not yet been linked with diabetic medical devices [64]. Clear labeling of the composition of device components would definitely help in further investigation of cross-reactions and co-reactivity between acrylates.

As stated previously, in a group of seven patients with skin reactions to FreeStyle Libre, all were sensitized to DMAA and six of them additionally to IBOA. The presence of both compounds was confirmed by the manufacturer, clearly pointing to co-reactions between IBOA and DMAA in these cases.

Recent studies have shown that 44% of patients with FreeStyle Libre-associated IBOA allergy have positive patch test results to sesquiterpene lactone mix (SLM) [38]. Interestingly, SLM has not been identified in FreeStyle Libre and IBOA patch test materials, nor has IBOA been demonstrated in any of the SLM patch test materials. Though the striking occurrence of concomitant patch test results towards IBOA and SLM requires further studies, the authors theorize that SLM can cross-react with IBOA [38,65]. A possible explanation for this phenomenon could be a common precursor for IBOA and SLM or (non)enzymatic reactions which, by triggering conformational changes, make IBOA mimic the α-methylene-γ- butyrolactone ring present in SLM responsible for cross-reactivity.

The summary of allergens causing ACD in patients using diabetes medical devices is presented in Table 1.

### 2.3. Irritant Contact Dermatitis (ICD)

ICD is caused by the direct toxic effect of an irritant compound which disrupts the skin barrier and triggers innate immune response with release of proinflammatory cytokines [66]. The first cases of dermatitis to medical devices were treated as ICD, since the repetitive occlusion, friction and increased humidity underneath adhesives are well-known irritants. On the other hand, damaged skin barriers are more permeable to allergens contained in the devices which, in turn, can lead to sensitization and possible evolution to ACD. Additionally, the coexistence of ACD and ICD is possible.

Reliable data concerning the prevalence of ICD in diabetic patients are not available. Herman et al. report that approximately 1/3 of patients experiencing adverse cutaneous effects from diabetes medical devices had no positive patch test results, suggesting ICD as a diagnosis of exclusion [13]. This number, however, could be overestimated as the lack of full labeling of ingredients contained in devices hampers the identification of allergens and irritants. The discovery of IBOA as the major culprit allergen altered the initial assumptions related to ICD and shifted the diagnostic process toward determining the already known and new allergens present in the devices. At the same time, the irritant potential of acrylates is undoubted; thus, the conception that some groups of patients might actually suffer from ICD rather than ACD cannot be neglected.

The overview of factors contributing to the development of contact dermatitis in diabetic patients is presented in Figure 1.

### 2.4. Systemic Contact Dermatitis

Another form of rarely observed contact dermatitis is systemic contact dermatitis (SCD) (also known as Baboon Syndrome) in which a patient first becomes cutaneously sensitized towards an allergen and upon a systemic re-exposure develops a sequela of systemic symptoms such as malaise, fatigue, fevers, vomiting and musculoskeletal disorders [67]. The pathophysiology of SCD is still poorly understood. Though previously SCD was thought to be a type I hypersensitivity reaction, the general consensus is now that SCD is a type IV hypersensitivity reaction [68]. So far, there are no reports linking the use of diabetes medical devices and SCD.

## 3. The Difficulties of Allergen Avoidance

The principal treatment of ACD is avoidance of culprit allergens, something that is not easy to implement in the case of GS or CSII users. In cases of ACD, relocation of the device to a different body region is not helpful. According to the most recent guidelines, children and young adults under 25 years old should use CGMs [69]. In comparison to finger prick self-measurements, CGMs are more effective in minimizing hypoglycemia incidents and are proven to reduce hemoglobin A1c (HbA1c), even in patients who are not on insulin, likely due to a positive impact on the patient’s lifestyle and behavioral changes [70,71,72]. In Poland, a patient must meet specific requirements to receive reimbursement for a diabetes medical device. Once the patient receives the GS/CSII, a change of the device ‘on demand’ is possible in most cases only at the patient’s cost, which, for a majority of them, is not affordable due to the high prices of diabetic devices. Moreover, bearing in mind cross-reactions between acrylates, we cannot guarantee the patient that the new GS/CSII, which in theory should be free from the culprit allergen, will not elicit contact dermatitis to another allergen. In cases of ICD, a possible way to reduce the symptoms of contact dermatitis could be to reduce the attachment time of the device to the skin. However, the number of insulin infusion sets or sensors a patient can obtain per month at reimbursed prices is strictly determined. Therefore, any additional use of infusion sets or sensors resulting from more frequent reapplications must be covered by the patients themselves, posing a financial challenge.

Since avoidance of allergens is in many cases difficult to implement or even impossible, patients try different barrier methods with varying success. These include barrier sprays, hydrocolloid dressings, BB kinesiotherapy tapes and others. However, barrier methods have substantial drawbacks: (i) applying extra protective layers underneath the medical device can result in false readings of glucose levels and/or inappropriate insulin infusions, (ii) the surface of the skin might not be dry enough to secure the adhesive and hence the device may detach after its application, (iii) sometimes barrier methods can elicit ICD or ACD and may lead to an exacerbation of dermatitis. Finally, it is generally recognized that acrylates (such as methyl methacrylate [MMA], 2-hydroxyethyl methacrylate [HEMA], triethylene glycol dimethacrylate [TEGDMA]) can penetrate protective barriers such as latex, vinyl and nitrile gloves [30,73,74]. This can possibly explain why users allergic to IBOA-containing devices do not experience the desired alleviation of skin symptoms upon using the skin barriers. Notably, occlusives on the skin might lead to even higher exposure to acrylates [31].

Practice shows that some patients who have changed GS from IBOA-containing FreeStyle Libre to ‘allergen-free’ Dexcom G6 or FreeStyle Libre 2 still experience contact dermatitis. It is postulated that either previous sensitization can somehow trigger contact dermatitis when using IBOA-free devices or that there are still allergens that have not yet been identified (personal observations M.C. and M.T.). Manufacturers usually do not provide a list of exact chemicals contained in devices or generally state that ‘polyacrylates’ are present without specification; sometimes we fail to obtain any information about the composition. Unfortunately, the ‘trial-error’ scheme is in some cases the only solution for patients who give up on their devices due to itching, pain or sleep deprivation. In acute contact dermatitis, topical corticosteroids (TCS) are the first-line treatment. As the use of TCS cannot be a long-term solution due to the side effects, patients can be recommended to use topical calcineurin inhibitors (tacrolimus; pimecrolimus) to control the subinflammatory process.

## 4. Conclusions

The identification of IBOA as the major contact allergen in diabetes medical devices was the praeludium to further research of other allergens that might be present in glucose sensors and insulin infusion sets. The management of diabetes device contact allergy is challenging, as avoidance of allergens is not always achievable and the ‘allergen-free’ equivalents can still elicit contact dermatitis due to the presence of untested allergens or via irritant, toxic pathways. As long as the precise chemical compositions of the medical devices are not officially disclosed by the manufacturers, the burden of contact dermatitis amongst patients using diabetes sets will be considerable.

## Figures and Tables

**Figure 1 ijms-24-10697-f001:**
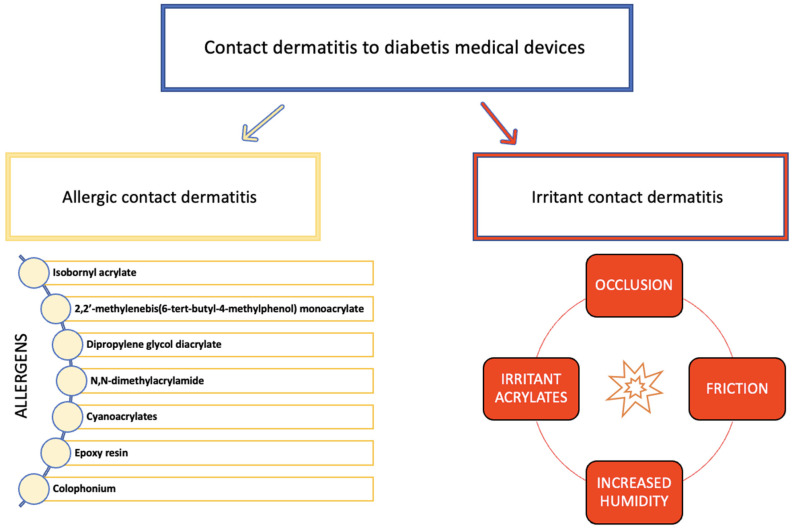
Factors contributing to the development of contact dermatitis from diabetes medical devices.

**Table 1 ijms-24-10697-t001:** Allergens responsible for contact dermatitis in diabetic patients using insulin infusion sets and/or glucose sensors. Only the allergens that patients were sensitized towards and whose presence in the devices was confirmed are listed.

Acrylate Allergens	Non-Acrylate Allergens
isobornyl acrylate (IBOA) [14,28,32,35,36,37,38,39,41]	colophonium [39,60]
2,2′-methylenebis(6-tert-butyl-4-methylphenol) monoacrylate (MBPA) [43,49]	epoxy resin [57]
dipropylene glycol diacrylate (DPGDA) [51]	nickel [61,62]
cyanoacrylates (e.g., methyl-2-cyanoacrylate and ethyl-2-cyanoacrylate) [39,52,53,54]	1-Benzoylcyclohexanol [32]
phenoxypoly(ethylenoxy) ethylacrylate (PEEA) [32,55]	
β-carboxyethyl acrylate (BCA) [32]	
N,N-dimethylacrylamide (DMAA) [36,56]	

## Data Availability

Not applicable.

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
