# Peer review of "Contact Dermatitis to Diabetes Medical Devices"

_ijms, 2023, doi:10.3390/ijms241310697_

Round 1

Reviewer 1 Report

This is a weel writen review article regarding contact dermatits among diabetic patients using a device. It is indeed a common problem for many patients using any kind of device attached to  theire skin. 

I Would recomend to discuss the question  whether we should recomend all patients before using a device to undergo patch testing. 

Author Response

Dear Reviewer,

Thank you for your time and your review.

We sincerely appreciate your comment.

The issue whether all patients should be patch tested prior to using a device is now discussed. [see lines: 193-210]

The English has been revised by a professional. 

With best regards 

Mikołaj Cichoń, M.D.

Department of Dermatology, Venereology and Allergology

Medical University of Gdansk

Reviewer 2 Report

The manuscript „Contact dermatitis to diabetes medical devices“ is a valuable manuscript which presents the most important data related with contact dermatitis to diabetes medical devices and their ingredients. Thus, in patients with diabetes mellitus, adhesives supporting glucose sensors and continuous insulin infusion sets to skin may cause contact dermatitis, including allergic contact dermatitis and irritant contact dermatitis. Therefore, several substances contained in adhesives or parts of the medical devices can cause allergic contact dermatitis, where acrylate chemicals (especially isobornyl acrylate; IBOA) are the most often culprit. In addition, some other factors such as repetitive occlusion, maceration of skin and following disruption of the skin barrier seem to have an impact on the development of skin lesions.  So, the authors highlight the burden of contact dermatitis triggered by diabetes medical devices and present possible mechanisms responsible for the development of contact dermatitis among diabetic patients, thus providing useful data in this topic.

-Abstract:  I suggest adding the names of more substances which may cause contact dermatitis to diabetes medical devices, as mentioned in the text.

-Text on pages 3 and 4 – I suggest dividing it into more paragraphs, to more clearly present the content.

-I think it would be needed to add more figures and/or tables. For example, it is possible to add tables, for instance a table with most common allergens and mentioned citing references. Thus, it could be very useful for the readers and clinicians (e.g. for further studies).  If possible, a clinical picture could be welcome.

-References same style./ Also, cited  references should be written in one uniform style

 -So, I think it is suitable for this journal, with the suggestions to check the text and style by an English professional.

Author Response

Dear Reviewer,

Thank you for your time and your review.

We sincerely appreciate your comments.

Referring to your comments and suggestions:

1. More substances which may cause contact dermatitis to diabetes medical devices were added to the abstract. [see lines 13-16]

2. Materials and Methods [see lines 61-75] and Risk factors [see lines 94-104] paragraphs were distinguished.  Moreover, paragraphs with difficulties in patch testing have been moved further down to the Allergic contact dermatitis section. [see lines 147-192]

3. A table with the most common allergens was added [see line 477]. We wanted to add an image of skin lesions that we observe in our patients. However, due to the publication policy of the journal we were provided with from the Editor, no original images can be embedded in a review article.

3. References are now written in an uniform style.

4. English language has been revised by a professional.

With best regards

Mikołaj Cichoń

Department of Dermatology, Venereology and Allergology

Medical University of Gdansk

Reviewer 3 Report

This is a well-written manuscript manuscript focusing on an interesting topic (contact dermatitis in patients with diabetes who use medical devices for their diseases). 

The introduction is well-written and provides enough background. 

In my opinion, the major issue with this work is the absence of a Materials and Methods section; being this a review (although not a systematic one),  it is fundamental to describe how literature search was performed. 

Moreover, i do not find appropriate for a scientific article to name a section "Finding the solution to contact dermatitis". I would rephrase it.

Minor revision of English language should also be performed 

The references are appropriate. No Ethical issues have been detected

I suggest just a minor revision of English language. No critical issued detected

Author Response

Dear Reviewer,

Thank you for your time and your review.

We sincerely appreciate your comments.

Referring to your comments and suggestions:

1. Although this is not a systematic review but a narrative review, we added Materials and Methods section according to your suggestion. However, we would like to pinpoint that this clinical problem is not widely covered yet and there is a limited number of reliable articles. [see lines 61-75].

2. The section “Finding the solution to contact dermatitis” has been rephrased to “The difficulties of allergen avoidance”. [see line 522]

3. English language has been revised by a professional.

With best regards

Mikołaj Cichoń, MD

Department of Dermatology, Venereology and Allergology

Medical University of Gdańsk

Reviewer 4 Report

Figure 1, Diabetis is incorrect, please changed by diabetes

no mention has been made of problems with the metal terminal of sensors or pumps.

Are there risk factors for the development of local skin reactions to the sensors?

no image to add to the manuscript?

Author Response

Dear Reviewer,

Thank you for your time and your review.

We sincerely appreciate your comments.

Referring to your comments and suggestions:

1. The name of the figure has been corrected.

2. We wanted to add an image of skin lesions that we observe in our patients. However, due to the publication policy of the journal we were provided with from the Editor, no original images can be embedded in a review article. A table with most common allergens was added to visualize the culprit allergens more clearly. [see line 477]

3. Nickel is reported as the culprit allergen responsible for ACD in two cases that were reported long time ago (1985 and 1998). We have not encountered other, more up to date reports of nickel (or other metal-induced) cases of ACD in this group of patients. Moreover, when we were enquiring the manufacturers about the composition of the devices we were informed that the devices are nickel free. Nonetheless, we will add this information to the article as it is a relevant information and might be helpful for healthcare providers in differential diagnosis of suspected allergens. [see lines 411-420]

4. The major risk factor for the development of local skin reactions to the sensors (and insulin infusion sets) is the use of any diabetes medical device in the past that, possibly, contained culprit allergens (like IBOA), regardless of presence of skin lesions. Use of sensors containing an allergen can be a source of sensitization and future skin reactions, although contact dermatitis might not be elicited yet. Upon change of the device some patients can experience contact dermatitis from allergens contained in the devices they used before. Also, atopic dermatitis history and resulting disruption in the epidermal barrier or other skin barrier abnormalities can facilitate the penetration of allergens and lower the inflammatory threshold, promoting both ACD and ICD. [see lines 94-104].

5. English language been revised by a professional.

With best regards 

Mikołaj Cichoń, MD

Department of Dermatology, Venereology and Allergology

Medical University of Gdańsk

Reviewer 5 Report

This review article entitled “Contact dermatitis to diabetes medical devices is informative and well written. It deserves to be accepted since there are no comments need to be addressed.

Author Response

Dear Reviewer,

Thank you for your time and your comments.

We sincerely appreciate your review.

With best regard on behalf of all authors,

Mikołaj Cichoń

Round 2

Reviewer 3 Report

Thank you for addressing all the comments

I think the manuscript has been greatly improved